# General Mental Health Is Associated with Gait Asymmetry

**DOI:** 10.3390/s19224908

**Published:** 2019-11-10

**Authors:** Hanatsu Nagano, Eri Sarashina, William Sparrow, Katsuyoshi Mizukami, Rezaul Begg

**Affiliations:** 1Institute for Health and Sport (IHeS), Victoria University, P.O. Box 14428, Melbourne, VIC 8001, Australia; hanatsu.nagano@vu.edu.au (H.N.); tony.sparrow@vu.edu.au (W.S.); 2Graduate School of Comprehensive Human Sciences, Faculty of Health and Sport Sciences, University of Tsukuba, Tsukuba, Ibaraki 305-8574, Japan; s1845203@s.tsukuba.ac.jp (E.S.); mizukami.katsuyos.ga@u.tsukuba.ac.jp (K.M.)

**Keywords:** motion capture, mental health, ageing, gait asymmetry, minimum foot clearance, falls prevention

## Abstract

Wearable sensors are being applied to real-world motion monitoring and the focus of this work is assessing health status and wellbeing. An extensive literature has documented the effects on gait control of impaired physical health, but in this project, the aim was to determine whether emotional states associated with older people’s mental health are also associated with walking mechanics. If confirmed, wearable sensors could be used to monitor affective responses. Lower limb gait mechanics of 126 healthy individuals (mean age 66.2 ± 8.38 years) were recorded using a high-speed 3D motion sensing system and they also completed a 12-item mental health status questionnaire (GHQ-12). Mean step width and minimum foot-ground clearance (MFC), indicative of tripping risk, were moderately correlated with GHQ-12. Ageing and variability (SD) of gait parameters were not significantly correlated with GHQ-12. GHQ-12 scores were, however, highly correlated with left-right gait control, indicating that greater gait symmetry was associated with better mental health. Maintaining good mental health with ageing may promote safer gait and wearable sensor technologies could be applied to gait asymmetry monitoring, possibly using a single inertial measurement unit attached to each shoe.

## 1. Introduction

The sensor technology application of interest to our research group is human locomotion and, specifically, how information from body-mounted sensors can be used to evaluate the characteristic limb motions of walking [1,2]. Almost all clinically-focused gait analyses concern the individual’s physical health; and gait mechanics, reflected in kinematic and kinetic measures of limb motion, are shown to be affected by lower limb injuries due to trauma and neurological and metabolic disorders. In contrast to most locomotion analyses, our aim here was to identify gait characteristics that can predict the emotional states associated with an individual’s mental health [3,4].

The general term ‘health’ encompasses many processes—physical, psychological, and social—but these effects interact and, consequently, contemporary lifestyles are creating a multitude of mental health concerns. Mental health problems are, therefore, increasingly prevalent [5] and contemporary health services planning requires an understanding of the role played by psychological factors. Extended use of electronic devices, for example, not only leads to physical discomfort associated with muscle fatigue but also psychological stress and depression [6,7]. It has been proposed, furthermore, that hormone secretion causes abnormal neurotransmission and these micro-level effects may lead to small but detectable changes in movement characteristics, which may not be visually observable [8,9]. High frequency data sampling and processing using sensor technology may, therefore, have the sensitivity to detect emotional influences on human motor functions [3]. The effects on gait of serious psychological problems and associated medications have previously been reported [10], but there has been no published evidence that general mental health is reflected in measurable changes to gait patterns.

Ageing is the primary factor in gait decline and falls-related injuries often lead to loss of healthy independent lifestyles [11]. A range of ageing-related health conditions related to balance loss, sensory deficits, and medication effects are known to cause significant deficits in locomotor control. In contrast, there is little evidence to show how psychological states affect gait control. The research question addressed in this report was, therefore, whether there is an association between mental health scales and locomotor function in community-dwelling senior citizens. If such links were demonstrated, it may be possible to use data from body-mounted sensors to detect changes to gait mechanics induced by affective states.

Slower walking is the most fundamental and readily perceived variable reflecting gait impairment; often associated with shorter, wider steps, and a prolonged double support time—the interval for which both feet are in contact with the ground [12,13]. In addition to these fundamental gait parameters, there has also been interest in impairments to foot trajectory control during the swing phase, the period in which the foot is out of contact with the walking surface. Research has focused on how reduced foot-ground clearance predisposes gait-impaired populations to a higher risk of tripping-related falls. The critical biomechanical variable associated with tripping is Minimum Foot Clearance (MFC), a biomechanical event approximately halfway through the gait cycle’s swing phase. At this time, the foot is very close to the walking surface (only 1–2 cm), foot horizontal velocity is maximum, and any destabilization to gait due to obstacle contact occurs during the less stable single-limb support phase.

Interestingly, while mean MFC is approximately equivalent in younger people, greater MFC variability is associated with a greater probability of tripping in older adults [14,15]. The central tendency and variability of the above gait cycle variables, in addition to MFC, reflects the capacity to maintain movement consistency across multiple stride cycles [16]. Research has shown that gait symmetry is a very useful concept in identifying harmonized, coordinated, bilateral lower limb control [12,17,18,19]. Physical impairment to one limb can cause gait asymmetry, but in this project, it was hypothesized that lower mental health may also disrupt the coordination of alternating limb movements, as measured using gait symmetry indices [17].

## 2. Materials and Methods

### 2.1. Participants

The participants were residents of Konosu City (Japan) aged over 50 years (*N* = 126; age 66.2 ± 8.38 years, height 1.55 ± 0.06 m and body mass 55.17 ± 8.38 kg, 96 females and 30 males). The sample size in this project was considerably greater than that usually recruited for similar gait biomechanics experiments involving individuals in the same age range. In most previous work concerning ageing effects on 3D motion capture gait data, participant samples were typically no more than 10 to 25 participants (young and older) [20]. All participants lived independently and were classified as healthy based on a medical self-assessment, with no reported locomotor or cognitive impairments. Participants were recruited via an advertisement circulated in the Konosu City council newsletter. Potential participants were screened by a registered nurse who also recorded their health status prior to the gait assessment. All participants undertook informed consent procedures mandated and approved by the University of Tsukuba research ethics committee. The data analyzed in the current study was based on a single assessment conducted on 7 June 2019.

### 2.2. Protocol

#### 2.2.1. Mental Health Assessment

The widely used General Health Questionnaire 12 (GHQ-12) was employed to measure the participants’ mental health. The GHQ-12 comprises of 12 questions, each answered using a 4-point scale, with responses ranging from ‘strongly agree’ to ‘strongly disagree’. When scored, the 4-point responses are re-categorised as either 0 or 1, with ‘strongly agree’ and ‘agree’ assigned 1 point and ‘strongly disagree’ and ‘disagree’ scoring 0. The total score is, therefore, in the range of 0 to 12, such that low scores reflect good mental health and higher scores less positive mental health.

#### 2.2.2. Gait Assessment

To model foot motion when walking, reflective markers were attached to the toe (defined anatomically as the superior surface of the most distal extremity of the foot) and the heel, the most proximal location on the foot [12]. Gait testing was then conducted on a 10-m walkway at preferred walking speed, with trials repeated until a minimum of 30 complete step cycles had been sampled. The reflective markers were sampled at 100 Hz using a three-dimensional (3D) camera system (Optitrack, NaturalPoint, Corvallis, OR, USA). To eliminate high frequency noise prior to further analysis, the raw 3D position-time data were smoothed using a low-pass Butterworth digital filter with a cut-off frequency of 6 Hz. Toe-off and foot-ground contact, i.e. heel-strike, were identified by applying conventional gait event algorithms using the heel and toe velocity and acceleration [19]. MFC was computed as the toe vertical local minimum within a time-sample window mid-way through the swing phase, using an in-house algorithm implemented in Visual3D (C-Motion, Inc.) script language [15]. Step length and width were defined as anterior-posterior and medio-lateral displacements between the heels at heel contact (Figure 1). Double support time was the temporal period when both feet were in contact with the walking surface, i.e. from heel contact to contralateral toe-off.

As illustrated in Figure 2, MFC was the vertical displacement of the toe from the walking surface during the mid-swing phase [21]. All gait data were described by mean ± Standard Deviation (*SD*) with the *SD* indicating intra-subject variability, reflecting the ability to maintain a consistent gait pattern across multiple step cycles.

A Symmetry Index (*SI*) was computed to indicate left-right differences in the selected gait variables [17]:(1)$$SI=\;\left|\frac{\left(R-L\right)}{\left(R+L\right)\times 0.5}\right|\times 100\left(\%\right)$$
where *R*/*L* indicates right/left foot leading (Figure 1). *SI* was applied into both mean and *SD* of each gait parameter.

### 2.3. Correlation Analysis

Pearson’s correlations were performed to determine the relationships between GHQ-12 scores and gait variables, step length, step width, double support time, and MFC. Correlations were applied to the variables’ mean, *SD*, *SI* mean, and *SI SD*. Multiple regression analysis (SPSS, Inc., Chicago, IL, USA) was also undertaken to investigate whether combinations of gait variables more strongly predicted GHQ-12 scores; this analysis therefore specified GHQ-12 as the dependent variable and gait variables as independent variables. Significant effects were confirmed when computed *p*-values were less than 0.05.

## 3. Results

Mean ± *SD* of GHQ-12 scores of the participants were 1.563 ± 2.080. In previous work using GHQ-12, respondents with mild to moderate mental health concerns were shown to have mean scores of 3 or higher [22,23]. Our participants would, therefore, be considered to have relatively good mental health. Table 1 summarizes the gait variable data. Gender effects were observed on MFC *SD* due to male participants showing higher variability (*p* < 0.01), while male participants were overall 2.3 years older (*p* = 0.033).

Pearson’s correlations between GHQ-12 and gait variables are presented in Table 2. The most notable findings are the strong positive correlations with all *SI* variables (*p* < 0.001), consistently indicating increased asymmetry with higher GHQ-12 scores, indicative of lower mental health. Furthermore, the variability of interlimb asymmetry, represented by the standard deviation of the *SI* scores, was also positively correlated with low mental health. Mean step width was moderately positively correlated with GHQ-12, while MFC was negatively correlated, indicating that lower mental health participants tended to walk with wider steps and lower foot-ground clearance. The *SD* of gait variables did not show significant correlations with GHQ-12 and age was not strongly associated with GHQ-12 (*p* = 0.263).

Multiple regression analysis indicated an overall *r* = 0.761/*r*^2^ = 0.580 when all gait parameters were entered into the regression model with GHQ-12 as the dependent variable. The regression model was determined to be statically appropriate for these data (F_17,108_ = 8.759, *p* < 0.001). Unstandardized coefficients further identified significant effects of independent variables as follows. An increase of one unit step width *SI* was predicted to increase GHQ-12 by 0.025 (*t* = 4.211, *p* < 0.001); step width *SD SI* adds 0.007 to GHQ-12 (*t* = 1.987, *p* = 0.049); double support time *SD SI* shows an incremental effect of 0.007 (*t* = 2.027, *p* = 0.045) and MFC *SD SI* was associated with a 0.013 increase (*t* = 3.339, *p* = 0.001). Obtained gait parameters in the current study thus indicated high predictability for GHQ-12 scores. Figure 3 plots the correlations between GHQ-12 scores and gait parameter asymmetry, excluding measurement errors to clarify the trend.

## 4. Discussion

Mental health has often been considered to negatively affect motivation and social engagement, leading to impaired physical capacity, including walking [24]. While psychological disorders have detrimental effects on walking performance and increased injury risk [10], it has not previously been demonstrated that older people’s mental health affects their gait. When data for all gait measures were entered into the multiple regression analysis, a strong association was found between mental health and gait function. Gait asymmetry was evident, with results revealing strong positive associations between the *SI* of all gait parameters (both mean and *SD*) and GHQ-12, such that lower mental health was associated with gait asymmetry. General mental health may, therefore, affect interlimb coordination across multiple step cycles (*SD SI*) leading to step width and double support variability. These variables have been linked to a reduced dynamic balance [12,16], suggesting that mental health concerns may increase the risk of balance loss. In addition, increased asymmetry in MFC *SD* was also found to be associated with general mental health, indicating an increased probability of tripping [15,21].

Increased gait asymmetry due to decline in general mental health may be due to one or a combination of the following: (i) impaired executive function [25,26]; (ii) reduced postural control [27], and (iii) disturbance to the adequate transport of neurological transmitters [28]. Although limited research is available to conclude that impaired executive function increases gait asymmetry, previous studies have reported that dual-tasking causes increased asymmetry in post-stroke individuals [29] and Parkinson’s patients [26]. Impaired walking as a result of dual-tasking predicts falls risks, in part, due to impaired executive functioning [30]. It is, therefore, reasonable to hypothesize that mental distress can impair cognitive executive functions leading to disturbances in coordinated bilateral limb movements. It has been also reported that mental health is reflected in reduced postural control when walking, reflected in less stable head and torso motion [31,32,33].

If confirmed in further research, the link between general mental health and gait control implies that maintaining good mental health with ageing may promote safer gait in older adults. As a future direction, wearable sensors such as inertial measurement units (IMUs), could be applied to monitor gait asymmetry, possibly using a single IMU attached to each shoe. Wearable sensors are being applied in gait measurement technologies for real-world applications and although there are challenges in obtaining accurate gait parameter estimations, advances in machine learning are increasing their viability [15]. Many of these studies have validated their sensor data using 3D motion capture as the gold standard. While wearable sensor systems using accelerometers and gyroscopes are still, by these criteria, relatively inaccurate, these devices could be sufficiently accurate to detect gait asymmetry. Furthermore, integration of foot-pressure measurement systems such as F-scan and Pedar into a wearable gait monitoring device could also detect asymmetric features in loading responses during foot-ground contact. Wearable sensor systems can also warn the user when gait variables, such as MFC, fail to attain a safe, pre-determined subject-specific criterion. Critical thresholds of other gait parameters, such as step length and width and double support time, could also be used in sensor-based falls prevention applications.

The current research concerning mental health and motor functions can be extended to determine the generalizability of the present results by including people from other city regions and having different characteristics with respect to employment, i.e. working or retired and socioeconomic status. While time constraints on gait testing in the current research only permitted toe and heel reflective markers, future work will be conducted to measure additional gait and balance parameters using a more comprehensive marker set.

## 5. Conclusions

Safe locomotion is a primary goal for senior adults’ active lifestyles. As described in this study, when older adults’ general mental health is negatively affected, walking patterns were found to become more asymmetrical; therefore, techniques should be employed to possibly reduce their risk of tripping and falling by enhancing their general mental state. Gait training combined with lower limb monitoring using wearable (e.g., shoe mounted) sensors could provide an effective population-based falls prevention intervention.

## Figures and Tables

**Figure 1 sensors-19-04908-f001:**
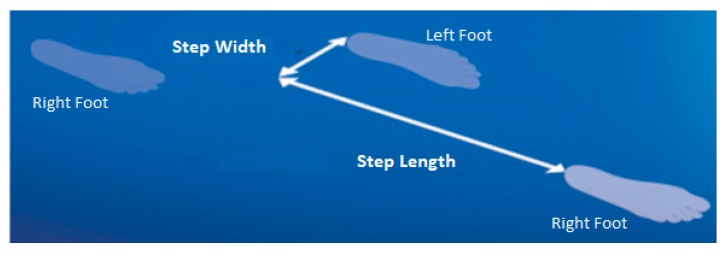
Right foot step length and step width represented by the displacement between the heels in the anterior-posterior and medio-lateral directions, respectively.

**Figure 2 sensors-19-04908-f002:**
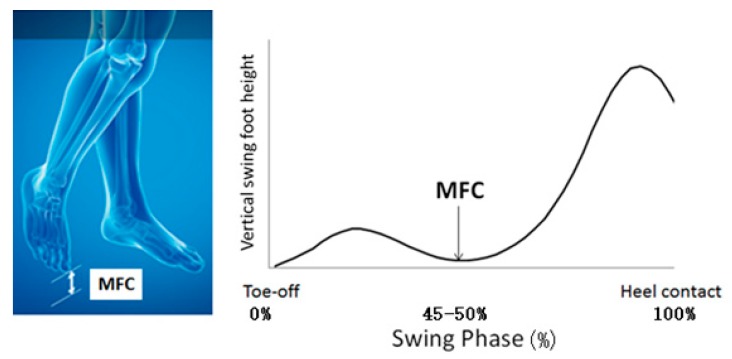
Minimum foot clearance (MFC), the lowest vertical height of the swing toe during mid-swing.

**Figure 3 sensors-19-04908-f003:**
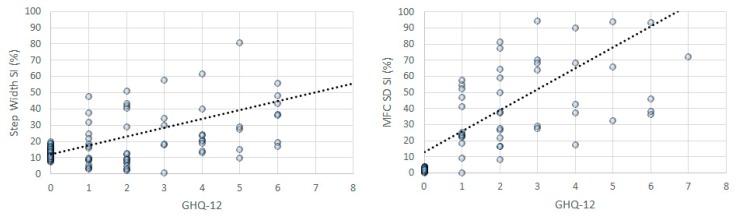
Scatter-plots for GHQ-12 and Gait Parameters. *X*-axis (scaling 0-8), *Y*-axis (scaling 0–100%).

**Table 1 sensors-19-04908-t001:** Data Summary: *SD* = mean intra-subject standard deviation, *SI* = symmetry index.

	Mean	SD	*SI* Mean	*SI SD*
Step Length	0.657 m	0.036 m	5.9%	35.4%
Step Width	0.121 m	0.030 m	14.8%	36.7%
Double Support	0.099 s	0.019 s	9.7%	33.9%
MFC	1.667 cm	447 cm	25.7%	32.4%

**Table 2 sensors-19-04908-t002:** Pearson’s correlations between GHQ-12 scores and gait parameters.

Correlations with GHQ-12	*r*	*p* Value
Step length mean	0.058	0.515
Step length (*SI*)	**0.366**	**<0.001**
Step length *SD*	−0.084	0.347
Step length *SD* (*SI*)	**0.401**	**<0.001**
Step width mean	**0.193**	**0.031**
Step width (*SI*)	**0.545**	**<0.001**
Step width *SD*	0.062	0.489
Step width *SD* (*SI*)	**0.537**	**<0.001**
Double support mean	−0.046	0.605
Double support (*SI*)	**0.436**	**<0.001**
Double support *SD*	−0.105	0.243
Double support *SD* (*SI*)	**0.480**	**<0.001**
MFC mean	**−0.179**	**0.045**
MFC (*SI*)	**0.379**	**<0.001**
MFC *SD*	−0.043	0.636
MFC *SD* (*SI*)	**0.545**	**<0.001**

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
