# Peer review of "General Mental Health Is Associated with Gait Asymmetry"

_sensors, 2019, doi:10.3390/s19224908_

Round 1

Reviewer 1 Report

The aim of this paper is to determine whether emotional states associated with older people’s mental health are associated with walking mechanics.

The paper can be of interest for some readers, but the following aspects must be clarified:

- inclusion and exclusion criteria of the subjects in this study must be presented;

- please specify if the number of participants is relevant for the presented study;

- in Section 2 the authors must show how the critical thresholds of the measured parameters (step length and width, double support time and MFC) can be determined, above which the measured values can be considered dangerous, triggering an alarm signal for the subject;

- in Section 4 (Discussion) the authors must focus on the experimental results in order to highlight the purpose of this study, the correlation between older people’s mental health and their gait;

- the conclusions of the study should be reviewed in order to focus on the results presented in this paper.

Author Response

inclusion and exclusion criteria of the subjects in this study must be presented

Some inclusion and exclusion criteria were described in the submitted manuscript (Lines 83-84) but further details have been provided as requested (Lines 87-91)

‘All participants lived independently and were classified as healthy based on a medical self-assessment, with no reported locomotor or cognitive impairments. Participants were recruited via an advertisement circulated in the Konosu City council newsletter. Potential participants were screened by a registered nurse who also recorded their health status prior to the gait assessment.’

please specify if the number of participants is relevant for the presented study

Justification has been provided for the sample size, showing that it was sufficiently large to detect gait effects using motion capture data from community-dwelling older adults (Line 83-85)

‘The sample size in this project was considerably greater than usually recruited for similar gait biomechanics experiments involving individuals in the same age range. In most previous work concerning ageing effects on 3D motion capture gait data, participant samples were typically no more than 10 to 25 young and older participants [20].’

in Section 2 the authors must show how the critical thresholds of the measured parameters (step length and width, double support time and MFC) can be determined, above which the measured values can be considered dangerous, triggering an alarm signal for the subject;

The Reviewer makes an important point in suggesting that critical thresholds of the measured parameters (step length and width, double support time and MFC) could be used in falls prevention applications. For example using wearable sensor systems to warn the user when gait variables, such as MFC, fail to attain a (safe) pre-determined subject-specific criterion. We have addressed the possibility of criterion-based warning systems in an extended discussion of the Reviewer’s comment, as follows: (Line 222-225)

‘Wearable sensor systems can also warn the user when gait variables, such as MFC, fail to attain a safe, pre-determined subject-specific criterion. Critical thresholds of other gait parameters, such as step length and width and double support time could also be used in sensor-based falls prevention applications.’

in Section 4 (Discussion) the authors must focus on the experimental results in order to highlight the purpose of this study, the correlation between older people’s mental health and their gait

Discussion section has developed the sufficient discussion on the highlighted results, association between ‘mental health’ and ‘gait’. (1st and 2nd paragraphs)

the conclusions of the study should be reviewed in order to focus on the results presented in this paper

 As suggested, the manuscript has been revised to highlight interlinks between mental state and gait asymmetry as below. (Line 233-236)

 ‘As described in this study, when older adults’ general mental health is negatively affected, walking patterns were found to become more asymmetrical; therefore techniques should be employed to reduce their risk of tripping and falling possibly by enhancing general mental state.’

Reviewer 2 Report

Please distinct the sex of the respondents and show all data accordingly.

Also, it is important to describe the study design. When the study was done? Please clearly state that.

What was the criteria for the classification of the respondents into the group without physical and cognitive impairment (medical doctor examination, individual medical documentation of each respondent etc.). Please describe.

How did you recruit your respondents? Was is convenient sample? Please explain.

What is the working status of the participants? In Europe the age for the retirement is 65 and I presume that within your sample there some retired and some people who are still employed. This difference is important for your study. Please explain.

Are these respondents representative for all the elderly people in Japan.

Was is it one time measurement or repeated measurements? Please clearly state.

Within the results section please show all of your data according to sexes (males and females separately) and according to age groups that you defined in the study.

Author Response

Also, it is important to describe the study design. When the study was done? Please clearly state that.

The information requested, has been provided in the revised manuscript. (Line 92-93)

‘Data analysed in the current study was based on single assessment conducted on 7th of June, 2019.’

What was the criteria for the classification of the respondents into the group without physical and cognitive impairment (medical doctor examination, individual medical documentation of each respondent etc.). Please describe.

As in the reply to Reviewer 1 – Comment 1, the following information has been added to the revised manuscript. (Line 87-91)

‘All participants lived independently and were classified as healthy based on a medical self-assessment, with no reported locomotor or cognitive impairments. Potential participants were screened by a registered nurse who also recorded their health status prior to the gait assessment.

How did you recruit your respondents? Was is convenient sample? Please

As in the response to Reviewer 1, the following sentence has been added to justify the reasonably large sample size of the current study. (Line 83-87)

‘The sample size in this project was considerably greater than usually recruited for similar gait biomechanics experiments involving individuals in the same age range. In most previous work concerning ageing effects on 3D motion capture gait data, participant samples were typically no more than 10 to 25 young and older participants [20]. Participants were recruited via an advertisement circulated in the Konosu City council newsletter.’

What is the working status of the participants? In Europe the age for the retirement is 65 and I presume that within your sample there some retired and some people who are still employed. This difference is important for your study. Please explain.

In the revised manuscript, the reviewer’s point has been incorporated as below. (Line 226-229)

‘The current research concerning mental health and motor functions can be extended to determine the generalisability of the present results by including people from other city regions and having different characteristics with respect to employment, i.e. working or retired and socioeconomic status.’

Are these respondents representative for all the elderly people in Japan.

Like most of other research studies involving human participants, our participants constituted a convenience sample, specifically individuals from the semi-rural region of Saitama prefecture approximately 150km north of Tokyo. This point has been acknowledged indirectly in reply above concerning participant characteristics.

Was is it one time measurement or repeated measurements? Please clearly state.

The current study focused on the results of one-time assessment as explained in the revised manuscript as follows. (Line 92-93)

‘Data analysed in the current study was based on single assessment conducted on 7th of June, 2019.’

Within the results section please show all of your data according to sexes (males and females separately) and according to age groups that you defined in the study.

The current research focus was the link between general mental health and walking function. Dividing the sample population further into sub-groups was not within the scope of the study due to consideration such as reduced sample size and unequal distribution of males and females. To address the reviewer’s comment, however, further information has been provided as follows.

‘Participants were residents of Konosu City (Japan) aged over 50 years (N = 126; age 66.2 ± 8.38 yrs, height 1.55 ± 0.06 m and body mass 55.17± 8.38 kg, 96 female and 30 male).’  (Line 82-83)

 ‘Gender effects were observed on MFC SD due to male participants showing higher variability (p < .01) while male participants were overall 2.3 years older (p = .033).’ (Line 153-155)

Reviewer 3 Report

In this work correlations were sought between general mental health and gait characteristics. A fairly large population (126) of older persons participated in the study. The data analysis is adequate and the results presented in a clear manner. The most interesting results is the strong correlation between the assymetry measure and mental health. However, there are some aspects of the results that merits more discussion (see below comments). The manuscript is well-written.

General comments

Figure 3 presents the most interesting results. However, it is stated that "extreme samples were excluded to clarify the trend". In my opinion all samples should be included to give a true picture of the data.  From figure 3 it also appears that if the data from the completely healthy individuals (GHQ = 0) would to be excluded, there would be only weak correlation between GHQ and the symmetry indices. Thus it seems that the symmetry index can differ between completely mentally healthy persons and persons with less positive mental health, but that between the levels of less positive mental health, there is little difference in the symmetry index. This aspect should be discussed in the article. It appears questionable that in the right-most plot of figure 3, that all individuals with GHQ=0 scored exactly 0% on the symmetry index MFC SD SI. There should be a thorough discussion and motivation for the particular gait measures chosen. Why not single-stance duration, gait speed? Limitations of the study should be discussed in the Discussion section.

Detailed comments

Line 90: Please provide references about GHQ-12 Line 127:  The formula should be SI= |R-L|/(0.5(R+L))*100

Author Response

However, it is stated that "extreme samples were excluded to clarify the trend". In my opinion all samples should be included to give a true picture of the data.

The description in this context was not accurate but ‘extreme samples were removed because they were considered to be measurement errors.’ (Line 82)

‘…with excluding measurement errors to clarify the trend.’

It appears questionable that in the right-most plot of figure 3, that all individuals with GHQ=0 scored exactly 0% on the symmetry index MFC SD SI. There should be a thorough discussion and motivation for the particular gait measures chosen. Why not single-stance duration, gait speed? Limitations of the study should be discussed in the Discussion section.

Figure 3 has been amended and the sentence below has been added to attend the reviewer’s concern about more data collection. (Line 226-231)

‘The current research concerning mental health and motor functions can be extended to determine the generalisability of the present results by including people from other city regions and having different characteristics with respect to employment, i.e. working or retired and socioeconomic status. While time constraints on gait testing in the current research only permitted toe and heel reflective markers, future work will be conducted to measure additional gait and balance parameters using a more comprehensive marker set.’

Line 90: Please provide references about GHQ-12 Line 127:  The formula should be SI= |R-L|/(0.5(R+L))*100 

In the revised manuscript, SI formula has been amended. (Line 134)

Round 2

Reviewer 1 Report

My suggestions and observations were taken into account.

My opinion is that the paper is of interest to some readers and can be accepted in this form.